# COVID-19 Vaccine Hesitancy and Associated Oral Cholera Vaccine Hesitancy in a Cholera-Endemic Country: A Community-Based Cross-Sectional Study in the Democratic Republic of Congo

**DOI:** 10.3390/vaccines12040444

**Published:** 2024-04-22

**Authors:** Arsene Daniel Nyalundja, Patrick Musole Bugeme, Alain Balola Ntaboba, Victoire Urbain Hatu’m, Guillaume Shamamba Ashuza, Jacques Lukenze Tamuzi, Duduzile Ndwandwe, Chinwe Iwu-Jaja, Charles Shey Wiysonge, Patrick D. M. C. Katoto

**Affiliations:** 1Center for Tropical Diseases and Global Health (CTDGH), Catholic University of Bukavu (UCB), Bukavu 285, Democratic Republic of the Congo; nyalu_arse.adn@outlook.com (A.D.N.); musolebugeme@gmail.com (P.M.B.); 2Faculty of Medicine, Catholic University of Bukavu (UCB), Bukavu 285, Democratic Republic of the Congo; balola.ntabobal1@gmail.com (A.B.N.); victoirehat@gmail.com (V.U.H.); shamambaguillaume@gmail.com (G.S.A.); 3Department of Epidemiology, Johns Hopkins Bloomberg School of Public Health, Johns Hopkins University, Baltimore, MD 21205, USA; 4Department of Global Health, Faculty of Medicine and Health Sciences, Stellenbosch University, Cape Town 7505, South Africa; drjacques.tamuzi@gmail.com; 5Cochrane South Africa, South African Medical Research Council, Cape Town 7501, South Africa; duduzile.ndwandwe@mrc.ac.za (D.N.); chinwelolo@gmail.com (C.I.-J.); sheyc@who.int (C.S.W.); 6Vaccine-Preventable Diseases Programme, World Health Organization Regional Office for Africa, Brazzaville P.O. Box 06, Congo; 7Centre for General Medicine and Global Health, Department of Medicine, University of Cape Town, Cape Town 7505, South Africa

**Keywords:** non-COVID-19 vaccine, *Vibrio cholerae*, South Kivu, immunization, vaccine misinformation

## Abstract

COVID-19 vaccine hesitancy and its enablers shape community uptake of non-covid vaccines such as the oral cholera vaccine (OCV) in the post-COVID-19 era. This study assessed the impact of COVID-19 vaccine hesitancy and its drivers on OCV hesitancy in a cholera-endemic region of the Democratic Republic of Congo. We conducted a community-based survey in Bukavu. The survey included demographics, intention to take OCV and COVID-19 vaccines, reasons for COVID-19 hesitancy, and thoughts and feelings about COVID-19 vaccines. Poisson regression analyses were performed. Of the 1708 respondents, 84.66% and 77.57% were hesitant to OCV alone and to both OCV and COVID-19, respectively. Hesitancy to COVID-19 vaccines rose OCV hesitancy by 12% (crude prevalence ratio, [cPR] = 1.12, 95%CI [1.03–1.21]). Independent predictors of OCV hesitancy were living in a semi-urban area (adjusted prevalence ratio [aPR] = 1.10, 95%CI [1.03–1.12]), religious refusal of vaccines (aPR = 1.06, 95%CI [1.02–1.12]), concerns about vaccine safety (aPR = 1.05, 95%CI [1.01–1.11]) and adverse effects (aPR = 1.06, 95%CI [1.01–1.12]), as well as poor vaccine literacy (aPR = 1.07, 95%CI [1.01–1.14]). Interestingly, the belief in COVID-19 vaccine effectiveness reduced OCV hesitancy by 24% (aPR = 0.76, 95%CI [0.62–0.93]). COVID-19 vaccine hesitancy and its drivers exhibited a significant domino effect on OCV uptake. Addressing vaccine hesitancy through community-based health literacy and trust-building interventions would likely improve the introduction of novel non-COVID-19 vaccines in the post-COVID-19 era.

## 1. Introduction

Cholera is a significant global health concern that threatens public health systems in low-resource settings, especially where the disease is commonly endemic [1]. The estimated burden of cholera accounts for 2.86 million (uncertainty range: 1.3–4.0 million) cases with 95,000 deaths (uncertainty range: 21,000–143,000) [2] and 1.9% of case fatality ratios (CFRs) [3]. While recent data suggest a decrease in cholera-related mortality globally between 1990 and 2019, significantly increasing mortality trends were observed in African regions [4]. In addition, Africa experiences the highest estimated CFR record of the decade at three times (2.9%) the agreed threshold (<1%) [5]. Global, multifaceted efforts have been undertaken to contain and control cholera, particularly in response to outbreaks, yet they remain insufficient to reduce the burden of disease related to cholera due to a combination of factors, including a lack of reliable data. Other challenges are not limited to complex humanitarian crises, political instability and protracted conflicts, health system fragility, climate change, limited and overstretched cholera workforce, multiple ongoing health emergencies, medical commodities supply chain, access to and availability of oral cholera vaccine (OCV), and more [1].

In the Democratic Republic of Congo (DRC), cholera is endemic in the eastern provinces, following an increasing seasonal trend during the rainy season [1,6,7]. The first cases of the disease were reported in 1973 during the seventh cholera pandemic, which began in Indonesia in 1961 [8]. Since the first cholera outbreak in 2021, the DRC reported 517,529 cases of cholera and 13,109 cholera-related deaths, ranking as the third and first country with the highest number of cholera cases and cholera-related deaths worldwide, respectively [9]. Sociopolitical instabilities and armed conflicts leading to massive internal and external displacement have been key factors in the cholera outbreak. For example, the most deadly and largest outbreak the country (more than 50,000 deaths) was observed in 1994 after the conflict in Rwanda [10]. Other factors, such as climate change and poor access to clean water, have been driving cholera across the country. In 2022, for example, a total of 18,403 suspected cases of cholera with a CFR of 1.6% were identified in 19 out of 26 provinces [6]. To control and eliminate cholera, the country implemented a nationwide strategy in 2007, “the Multisectoral Strategic Plan for Cholera Elimination” [11]. However, vaccination programs face enormous challenges including vaccine hesitancy [12], which is associated with vaccine safety and effectiveness concerns [13,14,15], trust in governments and scientists [16], complacency [15], knowledge gaps, poor health literacy, the infodemic and more. The government has received over 4 million doses targeting 2 million people. So far, over 1.4 million people have been vaccinated [17]. Although there are limited published studies on OCV uptake in the DRC, an OCV hesitancy rate of up to 67% has been recorded in a setting similar to the DRC [18]. However, mass vaccination against cholera provides herd immunity when more than 50% of the community receives two doses of a specific oral cholera vaccine [19].

Vaccine hesitancy, a highly variable and complex phenomenon, is patterned by specific contexts, time, and type of vaccines [20] and is classified by the World Health Organization (WHO) among the top ten threats to global health [21]. The COVID-19 pandemic further highlights this threat to global public health, especially in low-and-middle-income countries (LMICs), including the DRC, where vaccine-preventable diseases are the major contributors to the global burden of disease and where the COVID-19 vaccine uptake remained very low despite the availability of free vaccines. The DRC has one of the world’s lowest rates of COVID-19 vaccine coverage, with only 19% of the population receiving at least one dose by November 2023 [22]. This is likely due to high COVID-19 vaccine hesitancy.

The multifaceted drivers that have shaped the reluctance of LMIC communities to receive COVID-19 vaccines may play an important role in the willingness to receive novel non-COVID-19 vaccines in the COVID-19 endemic era. In this work, we hypothesized that COVID-19 vaccine hesitancy would increase the likelihood of the community in the DRC hesitating to use other novel vaccines, such as the OCV. In addition, community reasons for COVID-19 vaccine hesitancy and thoughts and feelings about COVID-19 and its vaccines would be associated with cholera vaccine hesitancy. These data might guide the expanded Programme on Immunization (EPI) to implement effective interventions accordingly.

## 2. Materials and Methods

### 2.1. Study Design Setting and Population

We conducted a household-based survey from 1 to 31 March 2022 in three sites in Bukavu, South Kivu. Bukavu is the capital city of the South Kivu province in the eastern DRC, and administratively, it has three municipalities: Ibanda (urban), Kadutu (peri-urban), and Bagira (peri-urban). The city is located southwest of Lake Kivu, and west of Cyagungu, Rwanda, from which it is separated by the Rizizi River. In 2022, it has an estimated 1,190,000 urban population [23]. All adults (aged 18 and above) in Bukavu formed the target population. In this study, the inclusion criteria required a respondent per household to be 18 years or older at the time of the survey and who had lived in the region for at least 12 months.

### 2.2. Sampling and Sample Size

A deliberate minimum interval of 15 households was established to facilitate the random selection of subsequent households. In every municipality, the selection of the respondents at the household level was based on convenience, adhering to the criteria of choosing one respondent from each household. For each study site, a minimum estimated sample size of 350 respondents was determined as per the WHO’s behavioral and social drivers of vaccines (BeSD) guidance [24].

### 2.3. Questionnaire

The survey contained closed-ended and Likert Scale questions adapted from the WHO’s BeSD. After it was piloted, the structured questionnaire was used in both the official language, French, as well as in the local language, Swahili. The structured questionnaire included three sections:Sociodemographic characteristics (7): age, gender, location, religion, religious acceptance of vaccination, educational level, profession, and monthly income;Cholera vaccine (3): respondents were asked if they have been vaccinated against cholera, if they were ready to receive this vaccine as soon as it became available, and if they were willing to have their children vaccinated against cholera;COVID-19 vaccine (4): vaccination against COVID-19, willingness to uptake COVID-19 vaccines if available, reasons for vaccination delayed or hesitancy, and perceptions about the COVID-19 vaccines.

Before administering the survey form, written informed consent was obtained for each respondent. Data were captured on tablets during the interview using KoboCollect (https://www.kobotoolbox.org/, accessed on 24 March 2023), an open-source Android application developed by KoboToolBox [24]. The survey was conducted by trained sixth-year medical students under the supervision of researchers affiliated with the Centre for Tropical Diseases and Global Health of the Catholic University of Bukavu.

### 2.4. Variables

The dependent variable was the willingness to uptake the cholera vaccine when available. The answers included “yes”, “no”, and “I do not know”. For this type of question, hesitancy toward the cholera vaccine was defined as “no” or “I do not know”, while “yes” was used to define vaccine acceptance. Independent variables were age, gender, location, religion, religious acceptance of vaccination, educational level, profession, monthly income, vaccination against COVID-19, willingness to uptake COVID-19 vaccines if available, reasons for delayed vaccination or hesitancy, and perceptions about the COVID-19 vaccines. Seven perceptions about the COVID-19 vaccines were identified, and each perception was designed as a 5-level Likert Scale question. The answers included “Strongly agree”, “Slightly agree”, “I am not sure/I have no opinion”, “Slightly disagree”, and “Strongly disagree”. To have binary variables, we merged “strongly agree” and “slightly agree” to become “yes” and the remainder to become “no”.

### 2.5. Data Analysis

We used R (The R Foundation for Statistical Computing, Vienna, Austria) version 4.2.2 for data cleaning and analysis. Data were summarized as counts and percentages for categorical variables and medians with an interquartile range (percentile 25 and percentile 75) for quantitative variables. Chi-square and Wilcoxon rank-sum tests were used as appropriate for group comparisons. We used modified Poisson regression to determine the incidence rate ratio (IRR), analogous to the prevalence ratio (PR). We then used the generalized linear model (glm) function in R to create four unique models to uncover factors independently related to respondents’ unwillingness to receive the cholera vaccine. Initially, the first model was adjusted to account for sociodemographic variables. Given the association between COVID-19 vaccine hesitancy and cholera vaccine hesitancy, the second and third models incorporated adjustments for variables related to the COVID-19 vaccine. The final model exclusively featured predictors that were significantly associated with vaccine hesitancy across the first three models, further adjusting for age as continuous variables and sex as binary variables, irrespective of their significance level. The results were presented as the PR with their 95% confidence interval (95%CI). All *p*-values were two-sided, and <0.05 indicated statistically significant results.

### 2.6. Ethics

This study was conducted in strict adherence to ethical principles. Access to the data was restricted solely to members of the research team, and any information that could potentially identify participants was meticulously removed or anonymized. Moreover, all participants provided informed consent for the publication of anonymized data. After a thorough explanation of any potential privacy risks, participants voluntarily agreed to the dissemination of findings in academic or professional forums. These measures were meticulously implemented to safeguard the privacy and confidentiality of all participants involved in the study.

## 3. Results

### 3.1. Sociodemographic Characteristics and Distribution of Cholera Vaccine Hesitancy among Respondents

A total of 1708 adults aged 38 years (median age, 95%CI 36–40 years) were surveyed in Bukavu, eastern DRC. Most of the respondents were males (54.34%), Christians (97.19%), aged between 25 and 39 years (33.14%), living in urban areas (54.75%), highly educated (62.59%), and employed (80.97%) but with a monthly income between USD 50 and USD 200 (40.20%). Overall, 84.66% of respondents were hesitant to receive OCV and were older than those who were willing to receive it when available (median age in years, 40 vs. 30, *p* < 0.001) (Table 1).

### 3.2. Socio-Demographic Factors Associated with Cholera Vaccine Hesitancy

The first modified Poisson regression model was built to assess the association between sociodemographic characteristics and hesitancy to receive OCV once it was available (Table 1). After adjusting for different variables in the models, we found that hesitancy toward receiving OCV was 6% and 10% higher among respondents whose religion does not accept vaccination (aPR = 1.06, 95%CI [1.02–1.12]) and those who lived in a semi-rural area (aPR = 1.10, 95%CI [1.03–1.19]), respectively.

### 3.3. Effect of COVID-19 Vaccine Hesitancy on Willingness to Vaccine for Cholera

In our initial analysis, we investigated the association between COVID-19 vaccination attitudes and willingness to be vaccinated against cholera. We noted that individuals hesitant to receive the COVID-19 vaccine were approximately 12% more likely to express hesitancy towards the cholera vaccine (cPR = 1.12, 95%CI [1.03–1.21]) (Table 2). Following this, through two separate Poisson regression models, we evaluated how specific reasons and perceptions related to COVID-19 vaccine hesitancy influenced the prevalence of hesitancy to receive the cholera vaccine. Importantly, respondents who perceived that the COVID-19 vaccine could contain other viruses, such as Ebola (adjusted PR (aPR) = 1.05, 95% CI [1.01–1.11]), or could have other harmful health effects (aPR = 1.06, 95% CI [1.01–1.12]), or those who indicated having a lack of information about the vaccine (aPR = 1.07, 95% CI [1.01–1.14]) were found to have a higher prevalence of hesitancy towards the cholera vaccine (Table 3). In contrast, those who recognized that COVID-19 could be prevented by vaccination showed a 5% reduction in the prevalence of vaccine hesitancy (aPR = 0.95, 95% CI [0.91–0.99]) (Table 4).

Additionally, in a comprehensive model that accounted for statistically significant factors from the earlier models and controlled for age and gender, we identified that age, geographic location, and specific concerns about the COVID-19 vaccine (for instance, the fear it might contain viruses like Ebola) independently contributed to a higher prevalence of delaying or refusing the cholera vaccine. Notably, the understanding that COVID-19 is preventable through vaccination remained a significant factor; individuals with this knowledge exhibited a notably lower prevalence of hesitancy towards receiving the OCV when available (aPR = 0.76, 95%CI [0.62–0.93]).

## 4. Discussion

In this household-based cross-sectional study conducted in Bukavu, eastern DRC, we investigated the interplay between sociodemographic characteristics, perceptions towards the COVID-19 vaccine, and the overarching willingness to engage in cholera vaccination initiatives. Overall, a pronounced hesitancy toward OCV was discerned, with a notable inclination among the older demographic. Specifically, the analysis highlighted a significant association between COVID-19 vaccine hesitancy and hesitancy towards OCV, particularly attributed to fears of vaccine contamination with viruses like Ebola, concerns about adverse health effects, and insufficient vaccine-related information. Further, an incremental hesitancy associated with religious opposition to vaccination and residence in semi-rural areas was observed. Strikingly, the perception that vaccines are effective in preventing COVID-19 corresponded with a lower hesitancy towards receiving OCV.

The associations observed between vaccine hesitancy and various factors in this study are likely reflective of broader trends beyond the DRC. For example, misinformation surrounding vaccine safety and efficacy has been a longstanding challenge across various vaccination campaigns, contributing to hesitancy. This study’s insights into the domino effect of COVID-19 vaccine hesitancy on OCV hesitancy highlight the long-term benefit of addressing misinformation and improving health literacy [25]. Importantly, while religious acceptance of vaccines did not significantly influence OCV hesitancy in our final model, it was linked to a notable rise in hesitancy in preliminary adjustments for sociodemographic characteristics. Additionally, previous studies conducted in similar settings indicated that OCV hesitancy was driven by religious reasons or tradition, along with community rumors regarding vaccine safety and confidence [26,27,28,29,30]. This highlights the influential role of community and religious leaders in vaccine uptake, corroborating literature that stresses their effectiveness in building trust and promoting health interventions over political or medical authorities [29,31].

Hence, the study further aligns with existing literature on the detrimental effects of public distrust in health institutions and government on vaccine hesitancy, exacerbated during the COVID-19 pandemic by the rapid spread of misinformation and conspiracy theories [32,33,34,35,36]. The emergence of an ‘infodemic’—an overload of both accurate and false information—has been identified as a significant barrier to trust in health services, potentially explaining the observed association between vaccine hesitancy and religious acceptance of vaccination [37,38,39,40]. Yet, notably, our data suggest that those recognizing the preventive efficacy of vaccines against COVID-19 exhibited less hesitancy towards OCV, emphasizing the importance of health education in dispelling myths and promoting vaccine acceptance [40].

The importance of this study is further highlighted by the ongoing health challenges in Africa, particularly the persistent and widespread cholera outbreak and the high mortality rates due to diseases such as malaria, which continues to be a leading cause of death among children under five. The introduction of new life-saving vaccines against malaria, along with vaccines for HPV to prevent cervical cancer, marks a critical juncture in the continent’s public health efforts. These vaccines represent a beacon of hope for reducing the disease burden and improving health outcomes. However, the effectiveness of these vaccination programs is significantly compromised by vaccine hesitancy, which will also constitute a barrier to achieving widespread vaccine coverage and the realization of their potential benefits. This study’s findings, while focused on oral cholera vaccine hesitancy, shed light on broader vaccine hesitancy issues that could impact the uptake of other new vaccines.

From a policy perspective, leveraging the lessons learned from the COVID-19 pandemic’s global and national response offers a strategic pathway for enhancing the Expanded Programme on Immunization (EPI) and overcoming challenges posed by vaccine hesitancy. The pandemic has emphasized the importance of building trust in health systems, fostering clear and effective communication, and engaging communities directly to encourage vaccine acceptance. These strategies are not only pivotal for COVID-19 vaccine uptake but are equally applicable to the rollout of new vaccines for malaria, which remains a leading cause of mortality among children under five in Africa, and HPV. Adapting strategies that have successfully increased COVID-19 vaccine acceptance in the context of EPI programs can provide a robust framework for addressing vaccine hesitancy. This approach, coupled with efforts to strengthen health systems and enhance health literacy, holds the promise of significantly improving vaccine coverage for malaria, HPV, and other vaccine-preventable diseases, paving the way for better health outcomes and a reduction in child mortality across the continent. The EPI in the DRC should capitalize on these insights to develop comprehensive guidelines targeting vaccine hesitancy. These guidelines should adopt a multifaceted approach, including community engagement and education, tailored messaging, equitable vaccine access, healthcare worker training, and robust surveillance systems. By integrating these components, the EPI can effectively promote vaccine acceptance and enhance immunization rates.

One of the primary weaknesses of this study is its reliance on self-reported data, which may introduce bias due to respondents’ potential reluctance to disclose true vaccine hesitancy attitudes or misunderstandings about the vaccines in question. Furthermore, the cross-sectional design limits our ability to establish causal relationships between the identified factors and vaccine hesitancy. Additionally, while the study provides valuable insights into vaccine hesitancy within the context of the DRC, the findings might not be fully generalizable to other settings or populations due to cultural, socioeconomic, and health system differences. Despite these limitations, the study’s strengths lie in its substantial sample size and the use of robust statistical methods to assess the factors associated with vaccine hesitancy. This approach has allowed for a nuanced understanding of the interplay between various determinants of hesitancy, offering critical insights that can inform targeted interventions to enhance vaccine uptake in similar contexts.

## 5. Conclusions

We found that addressing COVID-19 vaccination hesitancy and its drivers related to vaccine content, safety, and misinformation is critical to improving the uptake of new vaccines in the post-COVID era. This observation creates an opportunity for immunization programs that can leverage collaboration between academic and research institutions, civil societies, health system agencies, and government bodies to improve thinking and feeling about vaccines and vaccine-preventable diseases through community-related health literacy interventions.

## Figures and Tables

**Table 1 vaccines-12-00444-t001:** Sociodemographic characteristics and predictors (Model 1) of OCV hesitancy among respondents.

Variables		Cholera Vaccine Hesitancy						*p*	cPR	95% CI		*p*	aPR	95% CI		*p*
		Yes		No		Total				Lower	Upper			Lower	Upper	
		n	%	n	%	n	%									
Age	Median^$^	40 (27–55)		30 (24–49)		38 (26–54)		**<0.001 ***	1.02	1.01	1.03	**0.02**				
	>65	104	6.09	12	0.7	116	6.79	**<0.001**	Reference				Reference			
	18–24	234	13.70	76	4.45	310	18.15		0.84	0.77	0.92	**<0.01**	0.91	0.82	1.02	0.14
	25–39	474	27.75	92	5.39	566	33.14		0.93	0.86	1.01	0.11	0.98	0.89	1.07	0.68
	40–54	372	21.78	55	3.22	427	25.00		0.97	0.90	1.04	0.46	0.99	0.92	1.09	0.92
	55–65	262	15.34	27	1.58	289	16.92		1.01	0.94	1.09	0.75	1.03	0.94	1.12	0.46
Gender	Female	666	38.99	114	6.67	780	45.66	0.49	Reference				Reference			
	Male	780	45.67	148	8.67	928	54.34		1.02	0.98	1.06	0.45	0.99	0.95	1.03	0.69
Religion	Christian	1409	82.49	251	14.7	1660	97.19	0.20	Reference				**-**			
	Non-Christian	37	2.17	11	0.64	48	2.81		0.91	0.78	1.06	0.14				
Religious acceptance of vaccines	Yes	738	43.21	178	10.42	916	53.63	**<0.001**	Reference				Reference			
	No	708	41.45	84	4.92	792	46.37		1.11	1.07	1.15	**<0.01**	1.06	1.02	1.12	**<0.01**
Location	Urban	617	36.12	156	9.13	773	45.25	**<0.001**	Reference				Reference			
	Semi-rural	829	48.54	106	6.21	935	54.75		1.11	1.06	1.16	**<0.01**	1.10	1.03	1.19	**<0.01**
Education status	High	887	51.93	182	10.66	1069	62.59	**0.01**	Reference				Reference			
	Low	42	2.46	2	0.12	44	2.58		1.15	1.07	1.24	**0.04**	1.09	0.99	1.20	0.16
	Medium	527	30.27	78	4.57	605	34.84		1.04	1.01	1.09	**0.03**	1.01	0.98	1.06	0.32
Employment	Yes	1197	70.08	186	10.89	1383	80.97	**<0.001**	Reference				Reference			
	No	249	14.58	249	4.45	498	19.03		0.89	0.83	0.94	**<0.01**	0.96	0.91	1.03	0.18
Monthly income	>200	198	16.51	40	3.34	238	19.85	**<0.001**	Reference				Reference			
	<50	438	36.53	41	3.42	479	39.95		1.09	1.03	1.17	**<0.01**	1.09	0.98	1.21	0.06
	50–200	400	33.36	82	6.84	482	40.20		0.99	0.93	1.06	0.94	0.97	0.88	1.07	0.60

Median^$^ with interquartile range Q1–Q3: interquartile range; * Mann–Whitney U test; cPR: crude prevalence ratio; aPR: adjusted prevalence ratio. The bold for the number denotes a statistically significant *p*-value indicating notable significance within the data analysis.

**Table 2 vaccines-12-00444-t002:** Association between cholera vaccine hesitancy and COVID-19 vaccine hesitancy.

Variables		Cholera Vaccine Hesitancy						*p*	PR	95% CI		*p*
		Yes		No		Total				Lower	Upper	
		n	%	n	%	n	%					
COVID-19 vaccine hesitancy	No	133	8.70	37	2.42	150	9.94	**<0.01**	Reference			
	Yes	1186	77.57	173	11.31	1359	90.06		1.12	1.03	1.21	**<0.01**

The bold for the number denotes a statistically significant *p*-value indicating notable significance within the data analysis.

**Table 3 vaccines-12-00444-t003:** Association between reasons for COVID-19 vaccine hesitancy and OCV hesitancy (Model 2).

Reasons for COVID-19 Vaccine Hesitancy		Cholera Vaccine Hesitancy						*p*	cPR	95% CI		*p*	aPR	95% CI		*p*
		Yes		No		Total				Lower	Upper			Lower	Upper	
		n	%	n	%	n	%									
Vaccines could contain other virus such as Ebola, or more	No	948	62.95	162	10.76	1110	73.71	**0.03**	Reference				Reference			
	Yes	356	23.64	40	2.66	396	26.30		1.05	1.01	1.10	**0.01**	1.05	1.01	1.11	**0.04**
Fear of vaccine side effects	No	822	54.15	137	9.03	959	63.18	0.43	Reference				Reference			
	Yes	488	32.15	71	4.68	559	36.83		0.98	0.94	1.02	0.38	0.99	0.94	1.05	0.08
Inefficacity of the vaccine	No	1051	69.24	178	11.73	1229	80.97	0.08	Reference				Reference			
	Yes	259	17.06	30	1.98	289	19.04		1.04	1.01	1.10	0.04	1.03	0.98	1.10	0.29
Vaccine could have other harmful effects	No	1011	66.6	178	11.73	1189	78.33	**0.01**	Reference				Reference			
	Yes	299	19.70	30	1.98	329	21.68		1.07	1.02	1.11	**<0.001**	1.06	1.01	1.12	**0.03**
Do not know where to get this vaccine	No	1228	81.06	195	12.87	1423	93.93	0.80	Reference				Reference			
	Yes	78	5.15	14	0.92	92	6.07		0.97	0.89	1.07	0.59	0.98	0.88	1.09	0.79
Poor knowledge about the vaccine	No	118	73.65	189	12.45	307	86.10	**0.04**	Reference				Reference			
	Yes	192	12.65	19	1.25	211	13.90		2.36	2.04	2.744	**<0.001**	1.07	1.01	1.14	**0.03**

The bold for the number denotes a statistically significant *p*-value indicating notable significance within the data analysis.

**Table 4 vaccines-12-00444-t004:** Association between COVID-19 vaccine perception and OCV hesitancy (Model 3).

Perceptions toward COVID Vaccine		Cholera Vaccine Hesitancy						*p*	cPR	95% CI		*p*	aPR	95% CI		*p*
		Yes		No		Total				Lower	Upper			Lower	Upper	
		n	%	n	%	n	%									
COVID-19 is a serious threat	No	576	33.72	59	3.45	635	37.17	**<0.01**	Reference				Reference			
	Yes	870	50.94	203	11.89	1073	62.83		0.89	0.86	0.92	**<0.001**	0.96	0.91	1.01	0.11
COVID-19 can be prevented by vaccination	No	942	55.15	105	6.15	1047	61.30	**<0.01**	Reference				Reference			
	Yes	504	29.51	157	9.19	661	38.70		0.84	0.81	0.88	**<0.001**	0.95	0.91	0.99	**0.01**
The risks of COVID-19 disease are greater than the risks associated with its vaccine	No	843	49.36	106	6.21	949	55.57	**<0.01**	Reference				Reference			
	Yes	603	35.3	156	9.13	759	44.43		0.89	0.85	0.93	**<0.001**	0.99	0.97	1.02	0.81
The COVID-19 vaccines I have access to are safe	No	1136	66.51	152	8.9	1288	75.41	**<0.01**	Reference				Reference			
	Yes	310	18.15	110	6.44	420	24.59		0.84	0.79	0.89	**<0.001**	0.97	0.93	1.01	0.10
I believe that my government is capable of delivering the COVID-19 vaccine everywhere in my country, to everyone and equally	No	1055	61.77	148	8.67	1203	70.44	**<0.01**	Reference				Reference			
	Yes	391	22.89	114	6.67	505	29.56		0.88	0.83	0.93	**<0.001**	0.98	0.95	1.01	0.17
I trust the science behind the COVID-19 vaccine.	No	1054	61.71	135	7.90	1189	69.61	**<0.01**	Reference				Reference			
	Yes	392	22.95	127	7.44	519	30.39		0.85	0.81	0.89	**<0.001**	0.97	0.94	1.01	0.10
Trust in the government	No	1149	67.27	187	10.95	1336	78.22	**0.01**	Reference				Reference			
	Yes	297	11.39	75	4.39	372	15.78		0.92	0.88	0.98	**<0.001**	0.98	0.95	1.01	0.30

The bold for the number denotes a statistically significant *p*-value indicating notable significance within the data analysis.

## Data Availability

The data and analytical code on which this article is based are available on request from the corresponding author (P.D.M.C.K.) or the first author (A.D.N.). To approve a request, it must be justified from a methodological point of view and receive the consent of all authors. All requests can be made after publication of this manuscript with no end date.

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
