# Peer review of "COVID-19 Vaccine Hesitancy and Associated Oral Cholera Vaccine Hesitancy in a Cholera-Endemic Country: A Community-Based Cross-Sectional Study in the Democratic Republic of Congo"

_vaccines, 2024, doi:10.3390/vaccines12040444_

Round 1

Reviewer 1 Report

Comments and Suggestions for Authors

Although cholera has been largely eliminated in most industrialized countries it remains a significant cause of morbidity and mortality in some parts of Southern Africa, including Democratic Republic of Congo (DRC). Simultaneously, although numerous campaigns aiming at promotion of oral cholera vaccine (OCV) have been implemented in numerous countries, there is a high level of vaccine hesitancy and refusal that hinder vaccination intake. While the problem of vaccine hesitancy has always been there it has become even more important in the post-COVID era. Thus, the manuscript tackles important but somehow neglected topic of association between COVID-19 vaccine hesitancy and OCV hesitancy. The unquestionable strength of the study is that while there are several studies on the topic conducted in Zambia, Sudan, Zanzibar or Mozambique, still there is scarcity on OCV hesitancy in DRC. However, while I believe that the paper could be of interest to the reads of the Journal it requires  some improvements before it could be considered for publication in the Journal.

1. The Introduction section is well prepared and provides a clear description and epidemiological data from the region. However, since the paper aims at describing the association between COVID-19 vaccine hesitancy and OCV hesitancy a brief information on the COVID-19 pandemic and COVID vaccine hesitancy in DRC should be given (morbidity and mortality rates, vaccine uptake, the COVID-19 vaccine policy/strategy in DRC and its effects, main reasons for COVID-19 vaccine hesitancy).

2. While the Materials and Methods is well prepared and described clearly it misses Ethical Issues subsection. Additionally, although the Institutional Review Board Statement is provided in the statement section at the end of the manuscript it should be also mentioned in this section.

3. The Discussion section is a bit vague. It lacks reference to previous studies on OCV hesitancy conducted in other low-and-middle-income countries from the region, including Zambia, Sudan, Zanzibar or Mozambique. Thus, the discussion needs to be tailored and compared to other countris.

-- Massing LA, Aboubakar S, Blake A, Page A-L, Cohuet S, Ngandwe A, et al. Highly targeted cholera vaccination campaigns in urban setting are feasible: the experience in Kalemie, Democratic Republic of Congo. PLoS Negl Trop Dis. 2018;12(5):e0006369.

-- Peprah D, Palmer JJ, Rubin GJ, Abubakar A, Costa A, Martin S, et al. Perceptions of oral cholera vaccine and reasons for full, partial and non-acceptance during a humanitarian crisis in South Sudan. Vaccine. 2016;34(33):3823–7.

-- Schaetti C, Ali SM, Chaignat C-L, Khatib AM, Hutubessy R, Weiss MG. Improving Community Coverage of oral Cholera Mass Vaccination Campaigns: Lessons learned in Zanzibar. PLoS ONE. 2012;7(7):e41527.

-- Heyerdahl LW, Pugliese-Garcia M, Nkwemu S, Tembo T, Mwamba C, Demolis R, et al. “It depends how one understands it:” a qualitative study on differential uptake of oral cholera vaccine in three compounds in Lusaka, Zambia. BMC Infect Dis. 2019;19(1):421.

-- Démolis R, Botão C, Heyerdahl LW, Gessner BD, Cavailler P, Sinai C, Magaço A, Le Gargasson JB, Mengel M, Guillermet E. A rapid qualitative assessment of oral cholera vaccine anticipated acceptability in a context of resistance towards cholera intervention in Nampula, Mozambique. Vaccine. 20182;36(44):6497-6505. doi: 10.1016/j.vaccine.2017.10.087.

For review:

--Trolle, H., Forsberg, B., King, C. et al. A scoping review of facilitators and barriers influencing the implementation of surveillance and oral cholera vaccine interventions for cholera control in lower- and middle-income countries. BMC Public Health 23, 455 (2023). https://doi.org/10.1186/s12889-023-15326-2

4. While it is very likeable that the link between covid vaccine and OCV exists the authors should be more cautious, since there are many other reasons for OCV hesitancy, and while some of them are mentioned in this paper, vaccine hesitancy has been there long before the COVID-19 pandemic and is also related to vaccines in general, including HPV or MMR vaccines.

5. Finally, the paper would benefit from adding some recommendations suggesting possible guidelines that should be implemented in order to overcome the problem discussed in the manuscript.

Minor:

Line 306: „6. Patents” should be deleted.

All in all, while in my opinion the paper requires some elaboration on its theoretical part, I appreciate this research and believe that it may be of interest to reader of the Journal. I am also convinced that since the issues raised in the article may help to understand some social factors that underline the OCV hesitancy in the LMIC it may stimulate the discussion on the effective campaign aiming at improving OCV rates in the country. For all these reasons, after revision I recommend its publication.

Author Response

  1. The Introduction section is well prepared and provides a clear description and epidemiological data from the region. However, since the paper aims at describing the association between COVID-19 vaccine hesitancy and OCV hesitancy a brief information on the COVID-19 pandemic and COVID vaccine hesitancy in DRC should be given (morbidity and mortality rates, vaccine uptake, the COVID-19 vaccine policy/strategy in DRC and its effects, main reasons for COVID-19 vaccine hesitancy).

Reply

Thank you for this important comment, we have reviewed this section to reflect this change. Line (91-94)

  1. While the Materials and Methods is well prepared and described clearly it misses Ethical Issues subsection. Additionally, although the Institutional Review Board Statement is provided in the statement section at the end of the manuscript it should be also mentioned in this section.

Reply

Thank you for your comment, we have added the Ethics sub-section as suggested. (Line 174-183)

  1. The Discussion section is a bit vague. It lacks reference to previous studies on OCV hesitancy conducted in other low-and-middle-income countries from the region, including Zambia, Sudan, Zanzibar or Mozambique. Thus, the discussion needs to be tailored and compared to other countris.

-- Massing LA, Aboubakar S, Blake A, Page A-L, Cohuet S, Ngandwe A, et al. Highly targeted cholera vaccination campaigns in urban setting are feasible: the experience in Kalemie, Democratic Republic of Congo. PLoS Negl Trop Dis. 2018;12(5):e0006369.

-- Peprah D, Palmer JJ, Rubin GJ, Abubakar A, Costa A, Martin S, et al. Perceptions of oral cholera vaccine and reasons for full, partial and non-acceptance during a humanitarian crisis in South Sudan. Vaccine. 2016;34(33):3823–7.

-- Schaetti C, Ali SM, Chaignat C-L, Khatib AM, Hutubessy R, Weiss MG. Improving Community Coverage of oral Cholera Mass Vaccination Campaigns: Lessons learned in Zanzibar. PLoS ONE. 2012;7(7):e41527.

-- Heyerdahl LW, Pugliese-Garcia M, Nkwemu S, Tembo T, Mwamba C, Demolis R, et al. “It depends how one understands it:” a qualitative study on differential uptake of oral cholera vaccine in three compounds in Lusaka, Zambia. BMC Infect Dis. 2019;19(1):421.

-- Démolis R, Botão C, Heyerdahl LW, Gessner BD, Cavailler P, Sinai C, Magaço A, Le Gargasson JB, Mengel M, Guillermet E. A rapid qualitative assessment of oral cholera vaccine anticipated acceptability in a context of resistance towards cholera intervention in Nampula, Mozambique. Vaccine. 20182;36(44):6497-6505. doi: 10.1016/j.vaccine.2017.10.087.

For review:

--Trolle, H., Forsberg, B., King, C. et al. A scoping review of facilitators and barriers influencing the implementation of surveillance and oral cholera vaccine interventions for cholera control in lower- and middle-income countries. BMC Public Health 23, 455 (2023). https://doi.org/10.1186/s12889-023-15326-2

Reply

Thank you for your comment. We have made appropriate change. Line 246 – 248

  1. While it is very likeable that the link between covid vaccine and OCV exists the authors should be more cautious, since there are many other reasons for OCV hesitancy, and while some of them are mentioned in this paper, vaccine hesitancy has been there long before the COVID-19 pandemic and is also related to vaccines in general, including HPV or MMR vaccines.

Reply

Thank you. The new version of the manuscript now reflect this change.

  1. Finally, the paper would benefit from adding some recommendations suggesting possible guidelines that should be implemented in order to overcome the problem discussed in the manuscript.

Reply

This important comment has been considered, and addition have been made accordingly. Line 301 to 306

Minor:

Line 306: „6. Patents” should be deleted.

Reply

We have made edit to reflect this change. Thank you

Reviewer 2 Report

Comments and Suggestions for Authors

This topic is actual and interesting due to the high rate of vaccine hesitancy related to COVID-19 pandemic. The results of this study can be used for plan and implement intervention for improve the vaccine uptake in the developing countries.

Recommendations and notes

Introduction is fine.

Methodology: The definition of Incidence Rate Ratio (IRR) refers to the ratio of two different rates of incidence. This ratio is calculated basically from cohort studies. Prevalence Ratio (PR) refers to the ratio of two different rates of prevalence. So, they are not the same. The Incidence Rate Ratio (IRR) is analog with Prevalence Ratio (PR). Please correct it.

Results: Remove this sentence „This section may be divided by subheadings. It should provide a concise and precise description of the experimental results, their interpretation, as well as the experimental conclusions that can be drawn.”

In the titles of Tabel 2 and 3 use COVID-19 instead of covid-19.

In Table 3 use capital letter for the first letter in „vaccine could have...”.

Table 4. Please use the same formula for COVID-19.

In rows 215-217: I do not understand clearly that if you have controlled your model for age and gender, how could you identify age as an independent contributing factor regarding refusing cholera vaccination. Please check it.

Discussion:

Row 223: Please remove „of 1708 adults” from this sentence.

Author Response

Introduction is fine.

Methodology: The definition of Incidence Rate Ratio (IRR) refers to the ratio of two different rates of incidence. This ratio is calculated basically from cohort studies. Prevalence Ratio (PR) refers to the ratio of two different rates of prevalence. So, they are not the same. The Incidence Rate Ratio (IRR) is analog with Prevalence Ratio (PR). Please correct it.

Reply

Thank you for this valuable comment. We have made change accordingly. Line 163

In the titles of Tabel 2 and 3 use COVID-19 instead of covid-19.

Reply

Thank you for valuable comment. We have made change accordingly.

In Table 3 use capital letter for the first letter in „vaccine could have...”.

Reply

Thank you for valuable comment. We have made change accordingly.

Table 4. Please use the same formula for COVID-19.

Reply

Thank you for valuable comment. We have made change accordingly.

In rows 215-217: I do not understand clearly that if you have controlled your model for age and gender, how could you identify age as an independent contributing factor regarding refusing cholera vaccination. Please check it.

Reply

We appreciate the opportunity to clarify this aspect of our analysis. When adjusting our model for age and gender, we aimed to account for the potential confounding effects these variables might have on the relationship between other factors and cholera vaccination refusal. By including age and gender as covariates in the multivariate analysis, we were able to evaluate the independent effect of each variable on the outcome, while holding the other variable constant.

The identification of age as an independent contributing factor comes from this multivariate approach, which allows for the assessment of each factor's unique contribution to the outcome of interest. In other words, even after controlling for gender and other variables, age emerged as a significant predictor of vaccination refusal. This suggests that the effect of age on vaccination refusal is not merely a byproduct of its association with gender or other factors but represents a distinct influence.

Discussion:

Row 223: Please remove „of 1708 adults” from this sentence.

Reply

Thank for your comment. We made appropriate edit. Line 225

Reviewer 3 Report

Comments and Suggestions for Authors

Estimated Authors,

First of all, please accept our apologies for this late review.

In fact, I've read with great interest your original article, reporting on the Vaccine Hesitancy / Acceptance for COVID-19 and associated cholera vaccine from the DRC, and more specifically from a region affected by high rates of cholera epidemic. In this study, COVID-19 was affected by substantial VH, and the effector of this conditions were well defined. Interestingly, effectors for VH resulted in a significant domino effect on cholera vaccine. These results suggest that community based interventions aimed to improve the health literacy have the potential to improve vaccination rates not only for a single vaccine (in this case, COVID-19), but also for other medical intervention. This is of substantial interest, as it is similarly reported from other settings and other vaccination programs.

From my point of view, the present paper could be accepted after a minimal English editing (i.e. $ mark in table 1, some minor typos scattered across the main text), and some editing of Methods section. More precisely, please provide the calculation you did apply in order to obtain your minimum sample size.

Comments on the Quality of English Language

The English text only requires some editing for residual typos.

Author Response

From my point of view, the present paper could be accepted after a minimal English editing (i.e. $ mark in table 1, some minor typos scattered across the main text), and some editing of Methods section. More precisely, please provide the calculation you did apply in order to obtain your minimum sample size.

Reply

Thank you for your invaluable comments. Edits have been made to reflect these changes.